# A Novel Technology for Separating Copper, Lead and Zinc in Flotation Concentrate by Oxidizing Roasting and Leaching

**Qian Zhang** [1,2], **Qicheng Feng** [1,2,*], **Shuming Wen** [1,2], **Chuanfa Cui** [3] **and Junbo Liu** [1,2]

1.  Faculty of Land Resource Engineering, Kunming University of Science and Technology, Kunming 650093, China; zqian9865@163.com (Q.Z.); shmwen@126.com (S.W.); 18314563709@163.com (J.L.)
2.  State Key Laboratory of Complex Nonferrous Metal Resources Clean Utilization, Kunming University of Science and Technology, Kunming 650093, China
3.  Civil engineering institute, Zhengzhou Vocational University of Information and Technology, Zhengzhou 450000, China; ccf99999@126.com
*   Correspondence: fqc@kmust.edu.cn or fqckmust@126.com; Tel.: +86-159-8719-0563

**Abstract:** In this work, oxidizing roasting was combined with leaching to separate copper, lead, and zinc from a concentrate obtained by bulk flotation of a low-grade ore sourced from the Jiama mining area of Tibet. The flotation concentrate contained 7.79% Cu, 22.00% Pb, 4.81% Zn, 8.24% S, and 12.15% CaO; copper sulfide accounted for 76.97% of the copper, lead sulfide for 25.55% of the lead, and zinc sulfide for 67.66% of the zinc. After oxidizing roasting of the flotation concentrate, the S content in the roasting slag decreased to 0.22%, indicating that most sulfide in the concentrate was transformed to oxide, which was beneficial to leaching. The calcine was subjected to sulfuric acid leaching for separation of copper, lead, and zinc; i.e., copper and zinc were leached, and lead was retained in the residue. The optimum parameters of the leaching process were: a leaching temperature of 55 °C; sulfuric acid added at 828 kg/t calcine; a liquid:solid ratio of 3:1; and a leaching time of 1.5 h. Under these conditions, the extents of leaching of copper and zinc were 87.43% and 64.38%, respectively. Copper and zinc in the leaching solution could be further separated by electrowinning. The effects of leaching parameters on the extents of leaching of copper and zinc were further revealed by X-ray diffraction and scanning electron microscopy analysis.

**Keywords:** copper–lead–zinc of flotation concentrate; oxidizing roasting; leaching

## 1. Introduction

As the three most common nonferrous metals, copper, lead, and zinc are extensively applied in various industrial fields and all aspects of modern life, including use in the medical industry, alloys, electroplating, rubbers, galvanizing, and chemical engineering [1–4]. With rapid industrial and economic development, resource consumption is increasing and single-metal deposits are gradually decreasing. As high-grade ores decline and ores increase in complexity, development and utilization of complex low-grade refractory ores has become an inevitable trend [5]. In nature, there are few deposits of single sulfide ores. Oxidized ores, which are generally related to sulfide ores, comprise mainly silicate and carbonate minerals of copper, lead, and zinc [6].

Froth flotation is widely applied in mineral processing for the production of copper–lead–zinc concentrates [7,8]. Mixed ores are difficult to recover through beneficiation operations; hence, enormous amounts of these resources remain unexplored, are stacked in open-air mines, or drained into tailing reservoirs, resulting in massive waste [9]. Many researchers have adopted leaching or metallurgy

beneficiation processes to recover copper, lead, and zinc from such ores [10–15]. Gargul et al. investigated the leaching of lead and copper from flash smelting slag using citric acid; the results showed that this hydrometallurgical method could successfully replace the existing treatment of slag in an electric furnace and converter [16]. Pb-free solder paste containing Sn, Bi, and Cu was treated by ammonia leaching followed by hydrochloric acid leaching to successfully separate the target elements [17]. Xu et al. improved copper recovery from refractory copper oxide ores using high-gradient magnetic separation, followed by secondary grinding and then leaching; the results demonstrated effective recovery of copper [18]. Asadi et al. [19] leach zinc from a lead–zinc flotation tailings sample using ferric sulfate and sulfuric acid. The results showed that the degree of influence on zinc leaching of various experimental parameters was in the order: temperature > stirring > liquid /solid ratio > acid/ferric sulfate ratio > sulfuric acid concentration. Under optimum conditions, zinc recovery of 94.3% was achieved. Tkacova et al. [20] researched selective leaching of zinc from a complex Cu–Pb–Zn concentrate and demonstrated deportment of copper and zinc into the leach solution, while lead and iron remained in the insoluble residue. Selective recovery of zinc increased with increasing reaction surface.

The copper–lead–zinc ore found in the Jiama mining area of Tibet has a high mud content, a high degree of oxidation, and complex properties, so it is extremely difficult to separate copper, lead, and zinc using a single flotation technology. Therefore, it is necessary to develop an alternative process to achieve separation of these. In this work, copper, lead, and zinc in a flotation concentrate obtained by bulk flotation of low-grade ore could be effectively separated by an oxidizing roasting–leaching–electrowinning process. This process provides helpful technical guidance for the comprehensive utilization of copper, lead, and zinc from these ores.

## 2. Materials and Methods

The copper–lead–zinc mixed concentrate sample was obtained by flotation of raw ore from Jiama, Tibet. Multicomponent chemical analysis showed that the concentrate contained 7.79% Cu, 22.0% Pb, 4.81% Zn, and 8.24% S. Phase analysis showed that it contained large quantities of sulfides. Copper was mainly in the form of copper sulfide and accounted for 76.79% of total copper, while 22.39% of copper was present as free copper oxide. Lead was mainly in the form of carbonate, accounting for 53.86% of total lead; lead sulfide comprised 25.55% of total lead, and lead sulfate accounted for 5.27%. Zinc was mainly present (67.66%) as zinc sulfide, with 29.54% as zinc oxide. Detailed analyses of the concentrate are presented in Tables 1–3.

**Table 1.** Phase analysis of copper (mass %).

| Phase | Sulfate | Free Copper Oxide | Combined Copper Oxide | Sulfide and Others | Total Copper |
|---|---|---|---|---|---|
| Content | 0.05 | 0.85 | 0.91 | 6.05 | 7.86 |
| Occupancy | 0.63 | 10.81 | 11.58 | 76.97 | 100.00 |

**Table 2.** Phase analysis of lead (mass %).

| Phase | Sulfate | Carbonate | Sulfide | Plumbojarosite and Others | Total Lead |
|---|---|---|---|---|---|
| Content | 1.16 | 11.85 | 5.62 | 3.37 | 22.00 |
| Occupancy | 5.27 | 53.86 | 25.55 | 15.32 | 100.00 |

**Table 3.** Phase analysis of zinc (mass %).

| Phase | Zinc Sulfate | Oxide | Sulfide | Franklinite and Others | Total Zinc |
|---|---|---|---|---|---|
| Content | 0.07 | 1.48 | 3.39 | 0.053 | 5.01 |
| Occupancy | 1.40 | 29.54 | 67.66 | 1.06 | 100.00 |

Oxidizing roasting experiments were conducted in a resistance furnace (Model: SX-6-16, Changsha Kehui Furnace Technology Co., Ltd., Changsha, China), and 1000 g copper–lead–zinc mixed concentrate was taken to a corundum dry pot and put into the resistance furnace. The temperature of the resistance furnace was set to 1200 °C in an oxygen atmosphere. When the temperature of the resistance furnace rose to the set temperature, timing started. After roasting for 3 h, the oxygen supply was stopped, and the power supply of the resistance furnace was turned off. Meanwhile, the oxygen-passing tube was pulled out, and the calcined product was taken out for cooling, manual crushing, and grinding to −74 μm for leaching.

The concentrated sulfuric acid used in this study was of analytical grade. Pure deionized water was used for all experiments. The dissolution experiments were conducted in a laboratory apparatus (DF-II digital display collector type magnetic stirrer, Jintan Shenglan Instrument Manufacturing Co., Ltd., Jintan, China), comprising a 250 cm$^3$ wide-mouth beaker batch reactor (Yak Glass Instrument Co., Ltd., Chengdu, China) equipped with a condenser to prevent evaporation losses, a digitally controlled mechanical stirrer, a thermometer to measure the temperature, and a thermostatically controlled water bath for heating. For each experiment, 20 g roasted calcine was added into the reactor, followed by deionized water and concentrated sulfuric acid (Chengdu Kelon Chemical Reagent Factory, Chengdu, China). The slurry was stirred at constant temperature and allowed to react for specified times, after which it was removed from the reactor, and subjected to solid–liquid separation. The solid residue was filtered, dried, weighed, and subjected to chemical examination (Zinc was tested by atomic absorption spectroscopy (Shanghai Pu Analysis Instrument Co., Ltd., Shanghai, China), and copper and lead were determined by volumetric method). The leaching efficiency, used to evaluate the leaching performance, was calculated as follows:

$$y = (1 - \frac{M \times \beta}{Q \times \alpha}) \times 100$$

where Q is the mass of the sample before leaching, M is the mass of the leach residue, and α and β are the copper, lead, or zinc grade of the calcine before leaching and the leach residue, respectively.

## 3. Results and Discussion

### 3.1. Oxidizing Roasting Tests of the Flotation Concentrate

It was necessary to convert the sulfides present to oxides by oxidizing roasting before leaching. The mixed concentrate was roasted at 1200 °C. The resulting calcine was ground using a grinder, screened to −74 μm particle size, and well-mixed prior to use in the leaching tests. Multi-element chemical analysis of this leach feed is shown in Table 4, and its X-ray diffraction (XRD) pattern is shown in Figure 1. The leaching test sample contained 11.04% Cu, 25.40% Pb, 7.58% Zn, 10.10% CaO, and 0.22% S. The grades of copper, lead, and zinc in the calcine were higher than that those of the mixed flotation concentrate.

XRD analysis showed no sulfide peaks, which indicated that the sulfide minerals were converted to oxidized minerals after oxidizing roasting, which was advantageous for development of the leaching flowsheet.

**Table 4.** Multi-element chemical analysis of roast calcine (mass %).

| Element | Cu | Pb | Zn | S | CaO | MgO | Fe |
|---|---|---|---|---|---|---|---|
| Content | 11.04 | 25.40 | 7.58 | 0.22 | 10.10 | 2.27 | 17.32 |

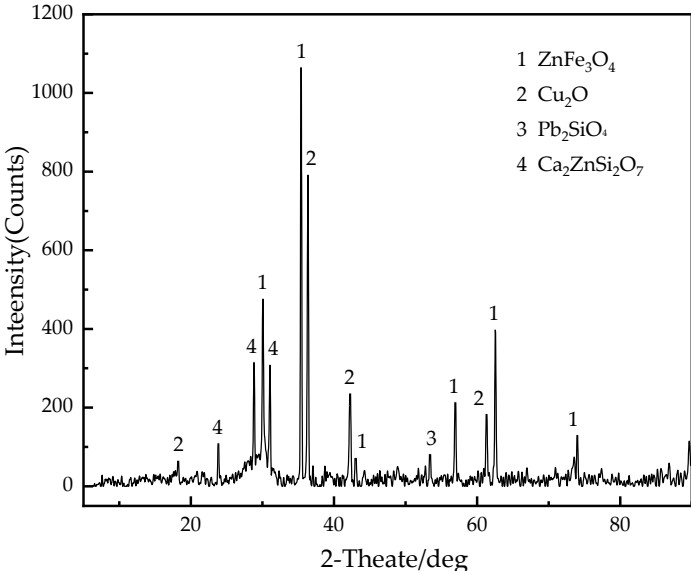

**Figure 1.** X-ray diffraction pattern of roast calcine.

### 3.2. The Leaching of Copper–Lead–Zinc

Leaching is one effective method for treating polymetallic ores, however, the effective leaching extent of metals is affected by many factors, such as temperature, leach reagent concentration, liquid:solid ratio, and leaching time [21,22]. Ye et al. [21] used bioleaching combined with brine leaching of heavy metals from lead–zinc mine tailings, which was a highly efficient method to recover lead, zinc, and other metals, but this technique required strict control of the slurry pH, and the reaction time was long with 5–30 days. Xu et al. [22] leached Cu and Zn in the mixture of calcine and soot by two-step countercurrent sulfuric acid leaching, and they studied the effect of leaching temperature, sulfuric acid concentration, liquid:solid ratio, and leaching time on the leaching rates.

#### 3.2.1. Effect of Leaching Temperature

Temperature is an important parameter affecting the leaching of copper and zinc in sulfuric acid solution. In each test, 20 g calcine was employed in the reaction beaker with a sulfuric acid dosage of 644 g/t at a liquid:solid ratio of 3:1. The leaching time was 1 h under constant temperature stirring at 25 °C, 35 °C, 45 °C, 55 °C, 65 °C, and 75 °C. The effect of reaction temperature on leaching is shown in Figure 2; XRD analyses of the leaching residues obtained at different temperatures are shown in Figure 3.

The grades of copper and zinc first increased and then decreased in the leach residue with the increase of temperature. Similar results were seen for their concentrations in the leaching solution. Copper leaching reached a maximum when the temperature was 65 °C, but leaching of zinc was lowest. A similar phenomenon was found when Yang et al. [23] studied the leaching of aluminum from secondary aluminum dross, and when Hyun et al. [24] investigated the leaching of CaO–SiO$_2$ resources. Considering the relative extents of copper and zinc leaching, the optimum temperature was selected as 55 °C. At this temperature, copper and zinc were leached to extents of 70.11% and 61.44%, respectively.

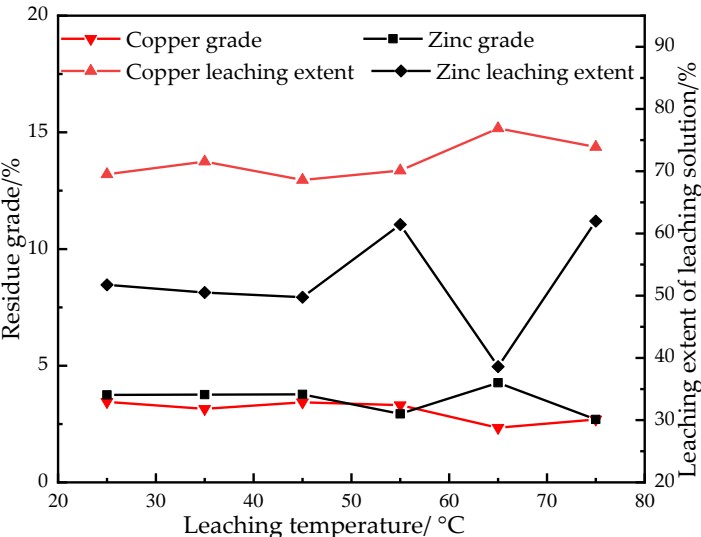

**Figure 2.** Results of leaching temperature tests.

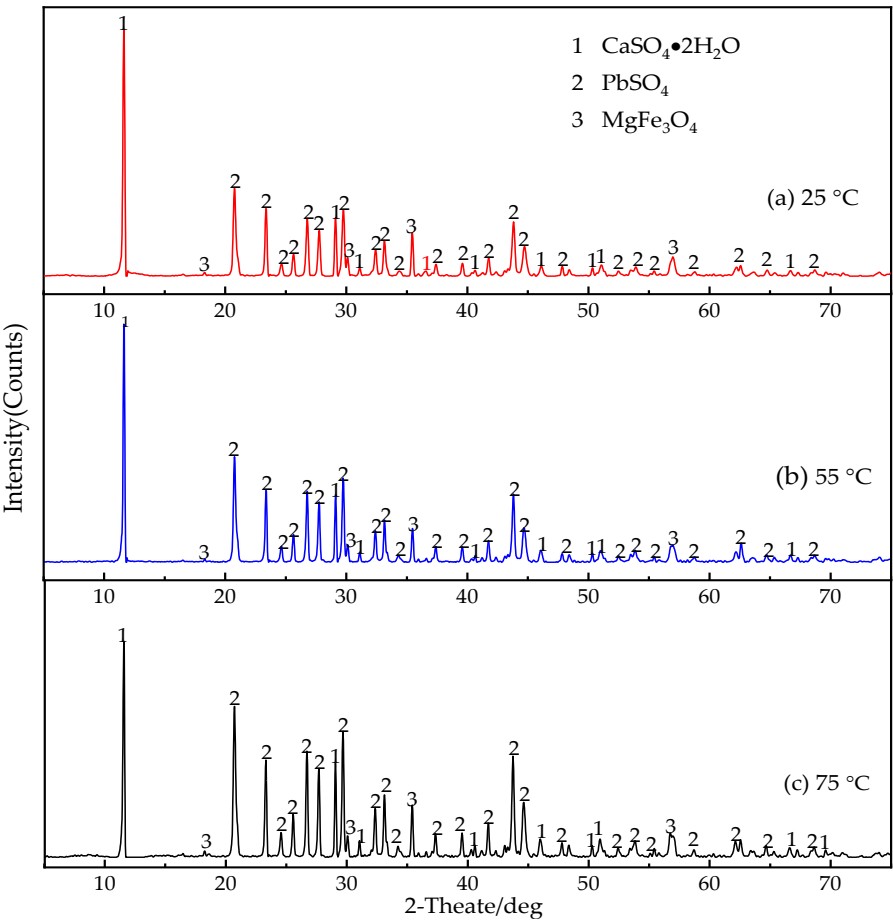

**Figure 3.** X-ray diffraction patterns of leach residues at different leach temperatures.

Comparing the XRD patterns of the solids before and after leaching at different temperatures, the composition changed obviously. The concentration of lead compounds in the residue increased, but no copper or zinc compounds appeared, indicating that most copper and zinc was leached. The compounds present in the calcine were the same at 25 °C, 55 °C, and 75 °C, but differences formed in the leach residue generated at different temperatures. When the temperature increased from 25 °C to 75 °C, the $PbSO_4$ content of the leach residue increased, reaching a maximum at 75 °C. Combined with

the leaching results, it was surmised that $PbSO_4$ may cover the surfaces of the copper and zinc minerals, resulting in a reduction of their leaching. This phenomenon was also reported by Kim et al. [10], who investigated the effect of temperature on the selective leaching of Pb and Cu from secondary lead smelting residues. They showed that the leaching extent of Pb and Cu dramatically decreased with temperature due to $PbSO_4$ formation at higher temperature.

### 3.2.2. Effect of Concentrated Sulfuric Acid Dosage

Dosage of the lixiviant is a very important factor affecting leaching. Experiments were conducted for 1 h to determine the effect of concentrated sulfuric acid dosage on 20 g calcine. The reaction temperature and liquid:solid ratio were kept at 55 °C and 3:1, respectively. Sulfuric acid dosages of 552 kg/t, 644 kg/t, 736 kg/t, 828 kg/t, and 920 kg/t were considered. The experimental results are shown in Figure 4; XRD analyses of the leach residues are shown in Figure 5.

Figure 4 shows that the grades of copper and zinc in the leach residue first decreased and then increased with the increase of sulfuric acid dosage; however, the solution analysis showed that leaching of copper and zinc first increased and then decreased. When the sulfuric acid dosage was 828 kg/t, the leaching of copper and zinc were maximized, at 89.04% and 63.86%, respectively. When the dosage of sulfuric acid was continuously increased to 920 kg/t, the leaching extent of copper and zinc minimized. This is attributed to the difficulty in filtration of leaching product because sulfuric acid reacted with $SiO_2$ in the leaching material to form the polymerization of silica sol, and the polymerization of silica sol increased with an increased acidity, which will clog the pores of filter paper, making filtration difficult [25]. Therefore, the optimum dosage of sulfuric acid is 828 kg/t.

XRD analyses of the leach residues obtained for different sulfuric acid dosages showed that the same compounds were present in the leach residue, but their concentrations differed with the increase of sulfuric acid dosage. When the dosage increased from 552 kg/t to 920 kg/t, the $PbSO_4$ content increased, reaching a maximum at 920 kg/t. It is postulated that too much $PbSO_4$ covered the surface of copper and zinc minerals and reduced the extents of their leaching. Combined with the results of the leaching test, it is considered that high sulfuric dosages were disadvantageous to the leaching of copper and zinc.

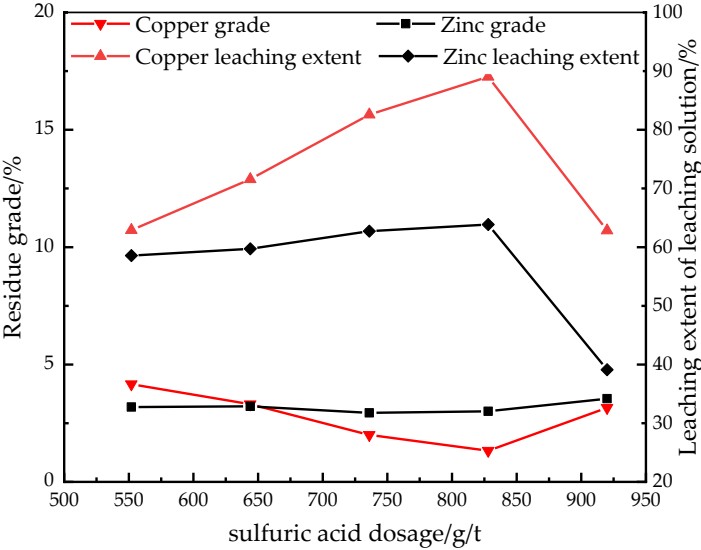

**Figure 4.** Results of leaching at different sulfuric acid dosage.

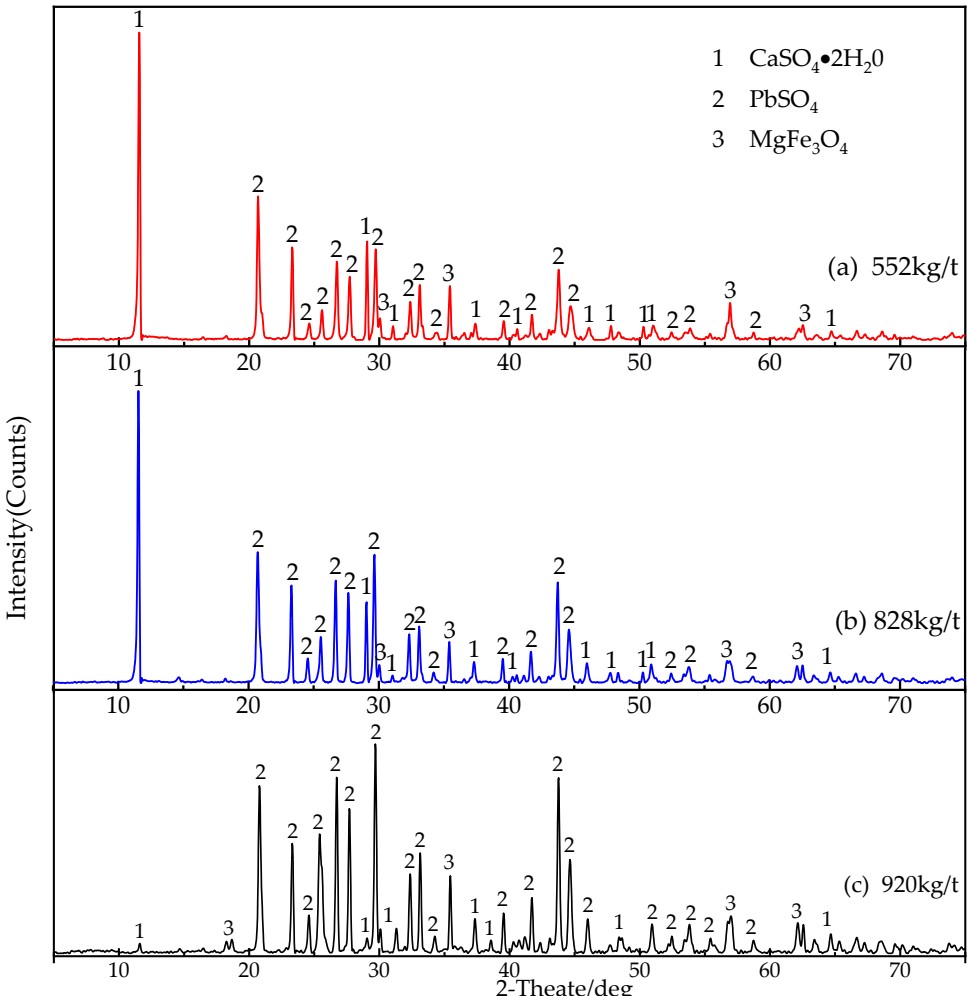

**Figure 5.** X-ray diffraction patterns of leach residues after leaching at different sulfuric acid dosages.

3.2.3. Effect of Liquid:Solid Ratio

Each test employed 20 g calcine at a leaching temperature of 55 °C and sulfuric acid dosage of 828 kg/t. Liquid:solid ratios of 2:1, 3:1, 4:1, and 5:1 were evaluated for a leaching time of 1 h. The experimental results are shown in Figure 6. XRD analyses of the leaching residues are shown in Figure 7.

The leaching of copper and zinc first increased and then decreased in the leach residue with increase of the liquid:solid ratio. The leaching of zinc was maximized at a liquid:solid ratio of 3:1. Leaching of copper was maximized at a liquid:solid ratio of 4:1, but that of zinc was at a minimum. Considering the relative extents of leaching, the optimal liquid:solid ratio was selected as 3:1, under which conditions the leaching extents of copper and zinc were 85.88% and 61.27%, respectively.

XRD analyses showed that the content of the $PbSO_4$ phase in the leach residue was larger when the liquid:solid ratio was small. As the liquid:solid ratio increased, the $PbSO_4$ content decreased, but a high liquid:solid ratio was disadvantageous to the leaching of metals; therefore, a liquid:solid ratio of 3:1 was considered appropriate.

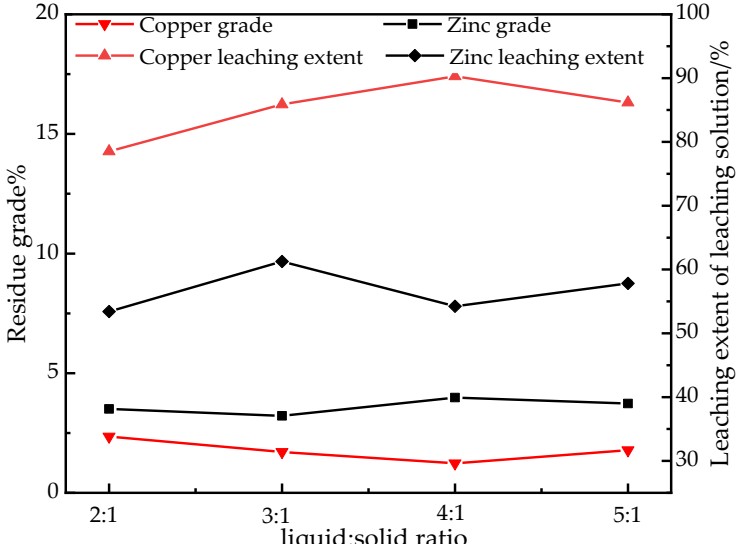

**Figure 6.** Results of leaching liquid:solid ratio tests.

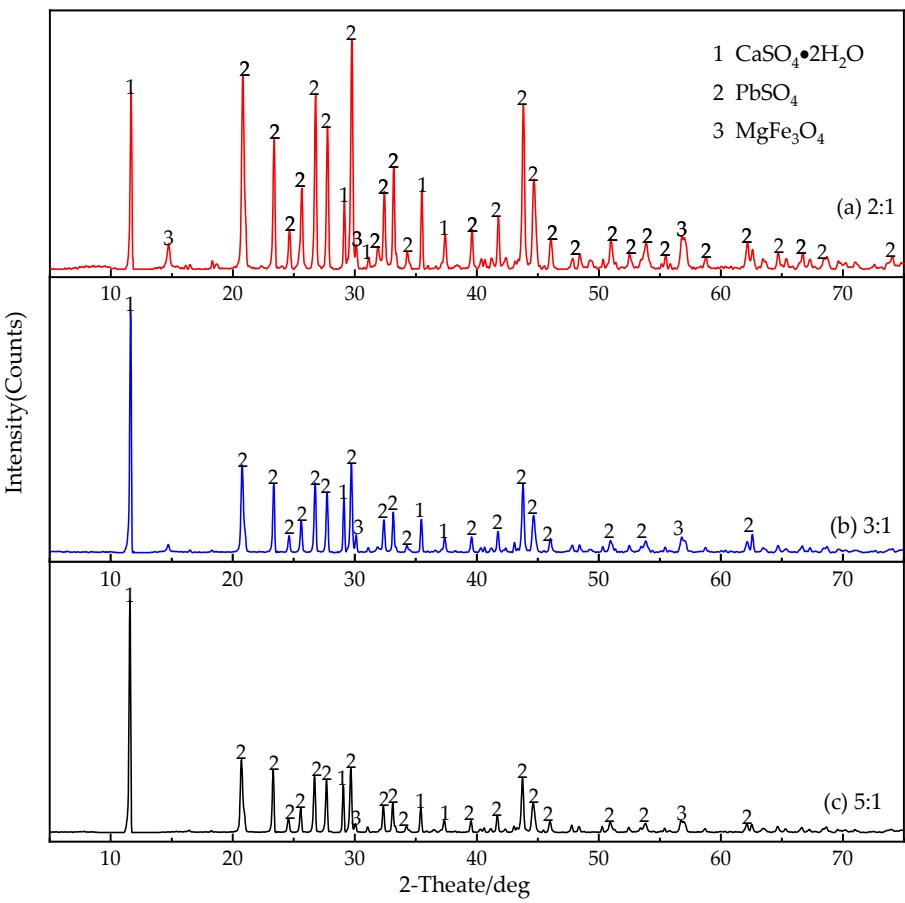

**Figure 7.** X-ray diffraction patterns of leach residue obtained at different liquid:solid ratios.

### 3.2.4. Effect of Leaching Time

The effect of leaching time was evaluated using 20 g roast calcine at a leaching temperature of 55 °C, sulfuric acid dosage of 828 kg/t, and liquid:solid ratio of 3:1 for times of 0.5 h, 1 h, 1.5 h, and 2 h. The results are shown in Figure 8; XRD analyses of the leach residue are shown in Figure 9.

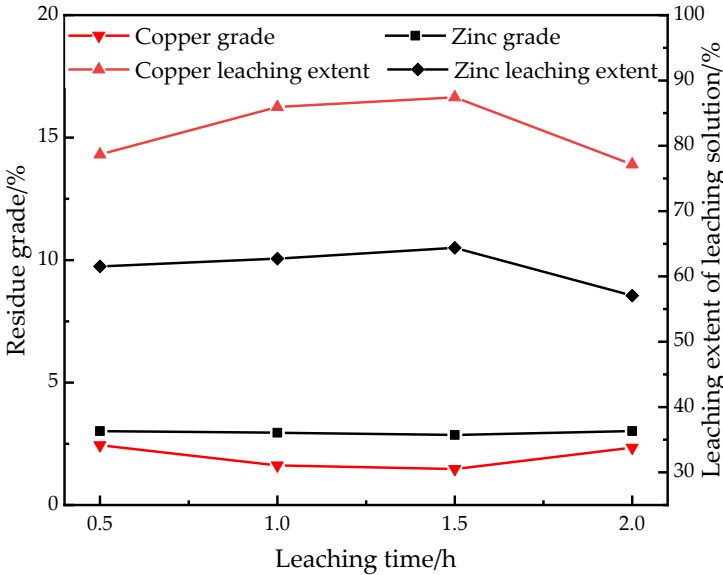

**Figure 8.** Results of leaching time test.

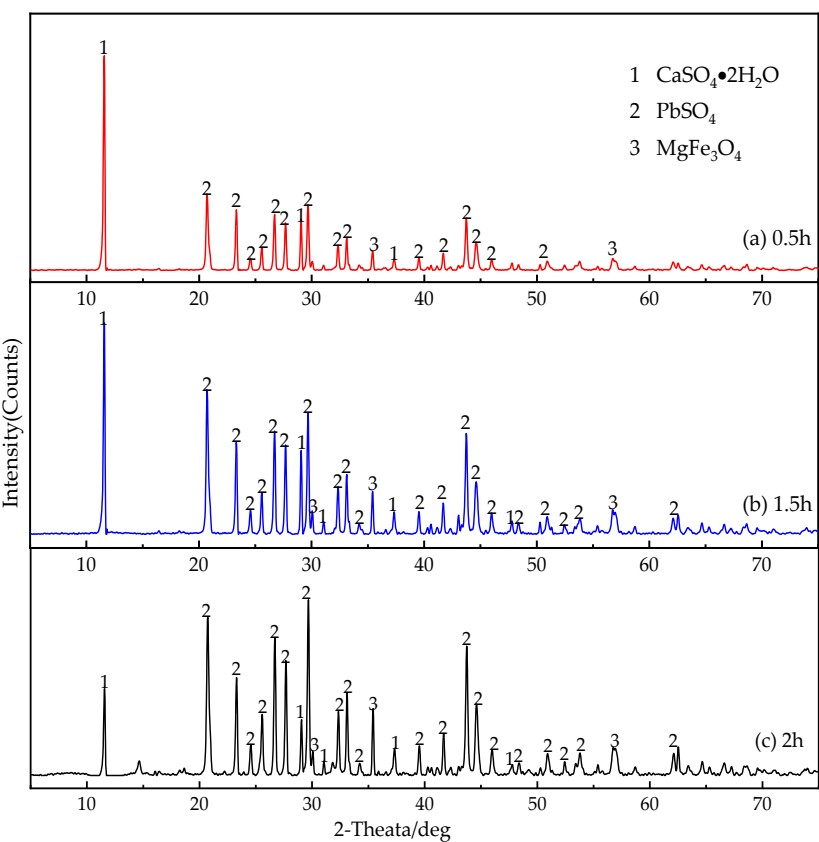

**Figure 9.** X-ray diffraction patterns of leach residues obtained for different leaching times.

The leaching extent of copper and zinc first increased in the leach residue, and then decreased with an increase of leaching time. The leaching of copper and zinc was maximized in the leaching solution at 87.43% and 64.38%, respectively, when leaching time was 1.5 h. This phenomenon was also reported by Zhang et al. [26], who investigated the effect of leaching time on Co leaching from laterite ore, and the results showed that the leaching extent of Co reached the maximum, and the leaching rate decreased, when the leaching time was prolonged.

XRD analyses showed that the PbSO$_4$ content increased with the increase of leaching time. When leaching for 2 h, the content of PbSO$_4$ increased to the maximum, but the leaching rate of copper and zinc decreased. It may be that a large amount of PbSO$_4$ covers the surface of copper and zinc, hindering the contact between leaching agent and target elements, and reduces the leaching rate of copper and zinc. Therefore, a leaching time of 1.5 h is more appropriate.

### 3.3. Scanning Electron Microscopy of Leach Solids Before and After Leaching

The leaching results and XRD analyses discussed in the preceding sections showed that copper and zinc could be effectively leached under certain conditions, and that the compositions and concentrations of the calcine feed and leach residue differed considerably. The morphologies of the roast calcine and leach residue were analyzed by SEM, as shown in Figures 10 and 11.

Comparing Figures 10 and 11 shows that significant changes occurred on the mineral surface during the leaching process. Before leaching, the surface of the calcine was smooth; after leaching, the structure of the calcine was destroyed and there were a lot of flocs on the surface. Combining these observations with the results of Figure 1, Figure 9b, it was concluded that the flocs may comprise CaSO$_4$ and PbSO$_4$ that covered the surface of the residue after leaching.

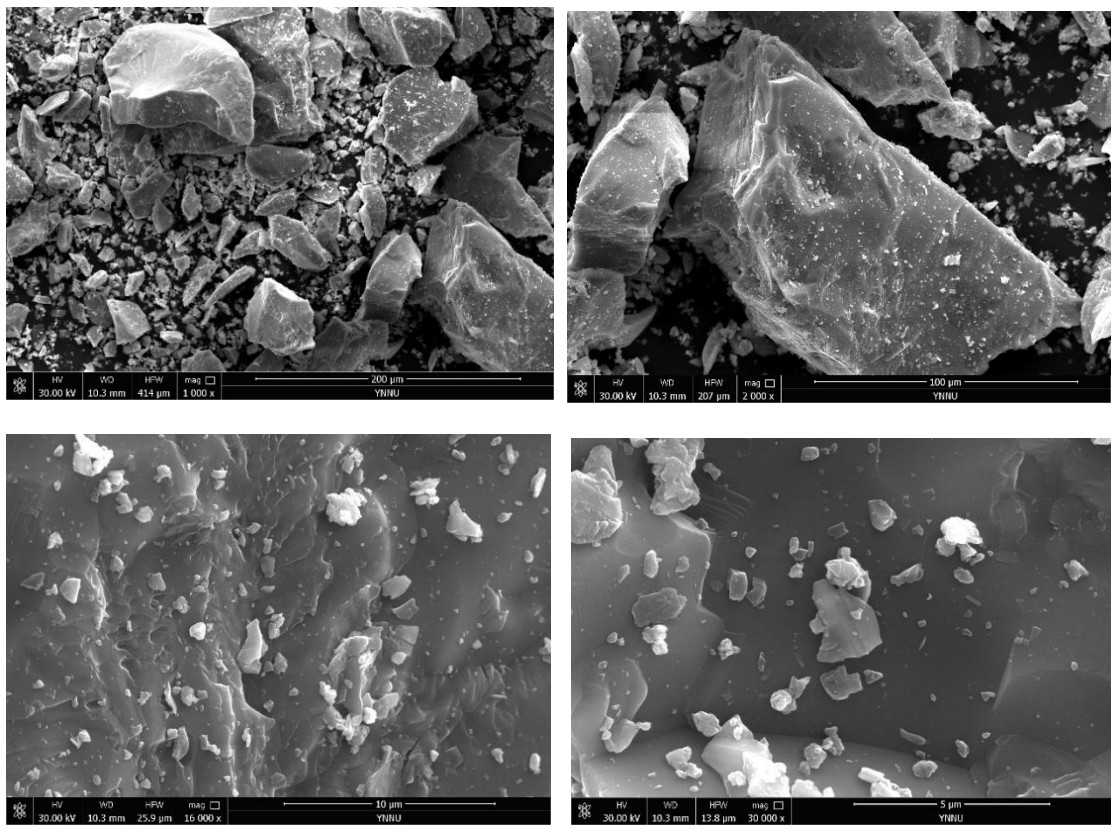

**Figure 10.** Scanning electron micrograph of morphology of roast calcine (before leaching).

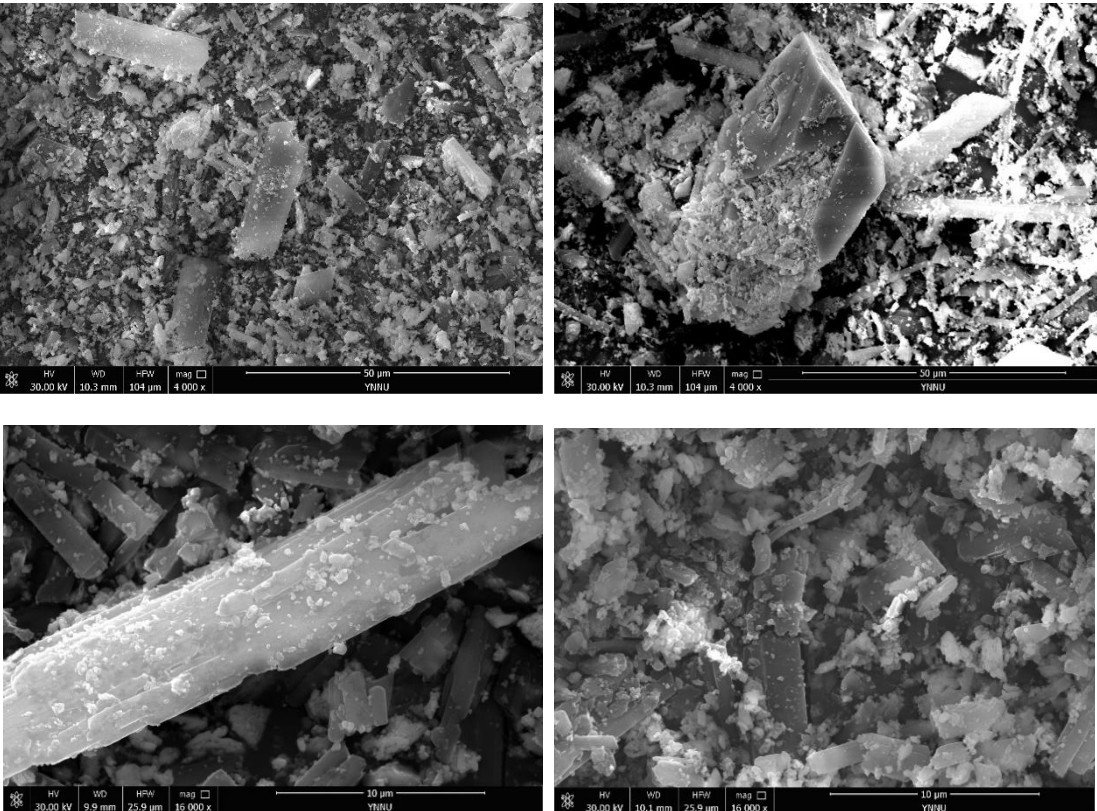

**Figure 11.** Scanning electron micrograph of morphology of roast calcine (after leaching).

## 4. Conclusions

Copper, lead, and zinc present in a flotation concentrate could be effectively separated by an oxidizing roasting–leaching–electrowinning process, which provides helpful technical guidance for the comprehensive utilization of ores from the Jiama mining area of Tibet.

(1) Sulfide in the flotation concentrate was transformed into oxide on oxidizing roasting at high temperature, which facilitated the subsequent leaching of copper and zinc in sulfuric acid solution. Copper and zinc in the roast calcine were leached to extents of 87.43% and 64.38%, respectively, at a leaching temperature of 55 °C, sulfuric acid dosage of 828 kg/t, liquid:solid ratio of 3:1, and leaching time of 1.5 h. It is proposed that copper and zinc in the leach solution could be further separated by electrowinning, and lead in the leach residue could be pyrometallurgy refined.

(2) XRD and SEM analyses indicated that the structure of the calcine was destroyed on subjection to sulfuric acid leaching, and its composition changed. Large amounts of $CaSO_4$ and $PbSO_4$ were produced during the leaching process and would covered the surface of the solid, which hindered the leaching of copper and zinc. Removal of calcium minerals prior to the sulfuric acid leach would be advantageous to the subsequent extraction of copper and zinc from calcine.

**Author Contributions:** Q.Z.; experimental, methodology, software, formal analysis, data curation, writing. Q.F.; resources, guidance review and editing. S.W.; investigate and guide. C.C.; data analysis. J.L.; experimental and data analysis.

**Acknowledgments:** This work was financially supported by Yunnan Applied Basic Research Project (Grant No. 2018FD035) and China Postdoctoral Science Foundation (Grant No. 2018T111000).

**Conflicts of Interest:** The authors declare no competing financial interest.

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
