# Peer review of "A Novel Technology for Separating Copper, Lead and Zinc in Flotation Concentrate by Oxidizing Roasting and Leaching"

_processes, doi:10.3390/pr7060376_

Reviewer 1 Report

This work has been suitably designed, and the results have been well presented. But it suffers from a poor discussion. All 20 references have been only cited in the Introduction section and there is no citation in the rest of the manuscript. The authors need to compare their results with what is already available in the literature (for comparison, and justification). 

Author Response

Comments and Suggestions for Authors: This work has been suitably designed, and the results have been well presented. But it suffers from a poor discussion. All 20 references have been only cited in the Introduction section and there is no citation in the rest of the manuscript. The authors need to compare their results with what is already available in the literature (for comparison, and justification).

Response to comments and Suggestions: Thanks very much for your approval of our research and precious suggestion for publication of our paper. On behalf of my co–authors, we would like to express our great appreciation to you because the rapid publication of the experimental results is very important for our further scientific research. The discussion has been supplemented in the revised manuscript according to the Reviewer’s comments, which was highlighted by red color in the revised manuscript.

Reviewer 2 Report

What is the novelty by this method?

You did regular leaching experiments, what is the advantage by this route?

Many reports already on leaching by sulfuric acid; what is your achievement by this method?

Write about econamic feasibility.

Author Response

Comments and Suggestions for Authors:

1. What is the novelty by this method?

Response to 1: Thank you for your careful review and this good question. Due to the complex nature of ore, copper, lead and zinc valuable metals are difficult to separate by flotation, and the price of copper-lead-zinc mixed concentrate obtained by bulk flotation is relatively low. In this work, oxidizing roasting was combined with leaching to separate copper, lead, and zinc from a concentrate obtained by bulk flotation of a low-grade ore sourced from the Jiama mining area of Tibet. The technology used in our work is novel, and not reported in other literature.

2. You did regular leaching experiments, what is the advantage by this route? Many reports already on leaching by sulfuric acid; what is your achievement by this method?

Response to 2: Thank you for your careful review and this good question. As mentioned in “Response to 1”, the technology used in our work is novel, and not reported in other literature. First, the copper-lead-zinc mixed concentrate obtained by bulk flotation was oxidizing roasted to remove S in the concentrate, i.e., after oxidizing roasting, most sulfide in the concentrate was transformed to oxide, which was beneficial to leaching. Through the leaching of sulfuric acid, the complex and difficult-to separate copper-lead-zinc mixed concentrate without economic benefit can be effectively separated, and the resource utilization rate and the economic benefits are improved. This process provides helpful technical guidance for the comprehensive utilization of copper, lead, and zinc from these ores.

3. Write about economic feasibility.

Response to 3: Thank you for your careful review and valuable suggestions. The flotation recovery of copper, lead and zinc in mixed concentrates was very low, and the copper-lead-zinc mixed concentrate was difficult to separate by using a single flotation technology, resulting in no economic benefit in the plant. Copper, lead and zinc in mixed concentrate was effectively separated after oxidizing roasting and leaching. The price of separated copper, lead and zinc is much higher than that of copper-lead-zinc mixed concentrate, which greatly increased the economic benefits and improved the comprehensive utilization of resources.

Once again, thanks very much for your comments and suggestions. We appreciate for your warm work earnestly, and hope that the correction will meet with approval.

Reviewer 3 Report

The manuscript processes-523041 covers an interesting topic of copper, lead and zinc recovery from polymetallic concentrates. It descibes a hybrid pyro- and hydrometallurgical methods for Cu, Zn and Pb separation from post-flotation concentrates from Jiama mining area in Tibet.

After careful reading of the content I cannot recommend to publish the manuscript in the current form. Please find below my remarks and comment the the manuscript.

Materials and Methods

1) More detailed description of the methods used should be provided. It should contain methodology of the whole investigation as well as all the equipment used for roasting, hydrometallurgy and chemical analysis. Suppliers of the equiment and materials should be provided. Moreover, I would recommend to move the description of oxide roasting experiment to section 3 (Results and discussion). Additionally, more detailed description of pyrometallurgical part could be provided.

2) What was the composition of raw ore?

3) What was the mass balance of the oxidizing roasting process? Was there any volatile content which was lost in the form of gas/dust

Results and Discussion

1) Temperature effect - Could you explain why zinc leaching extend/yield at 65oC was significantly lower than at other temperatures? Could you provide the value of the measurement uncertainity/error, i.e. in the form of error bars?

2) Different resolutions of XRD patterns in Figure 1 and 3 make comparision difficult. 

3) H2SO4 dosage effect - Could you explain why leaching extend of Cu and Zn significantly decrease when H2SO4 dosage is increased from 828 to 920 kg/t. Could you provide the concentration of Zn and Cu in the solution after leaching? Why you provide the concentration of H2SO4 in kg/t. It is usually presented in wt.% or M (molar) in the scientific publications.

Others

English need to be improved within the text. There are some mistypes, e.g. in the all Figures Znic should be replaced by Zinc, (line 105) Twenty grams of calcine were placed, etc.

Author Response

Comments and Suggestions for Authors:

The manuscript processes-523041 covers an interesting topic of copper, lead and zinc recovery from polymetallic concentrates. It describes a hybrid pyro- and hydrometallurgical methods for Cu, Zn and Pb separation from post-flotation concentrates from Jiama mining area in Tibet.

After careful reading of the content I cannot recommend to publish the manuscript in the current form. Please find below my remarks and comment the manuscript.

1. Materials and Methods

1) More detailed description of the methods used should be provided. It should contain methodology of the whole investigation as well as all the equipment used for roasting, hydrometallurgy and chemical analysis. Suppliers of the equipment and materials should be provided. Moreover, I would recommend to move the description of oxide roasting experiment to section 3 (Results and discussion). Additionally, more detailed description of pyrometallurgical part could be provided.

Response to 1): Thank you for this good suggestion. We have added the detailed description of the methods, equipment used for roasting, hydrometallurgy and chemical analysis, and suppliers of the equipment and materials according to the Reviewer’s suggestion, and the following expression was presented in the revised manuscript.

“Oxidizing roasting experiments were conducted in a resistance furnace (Model: SX-6-16, Changsha Kehui Furnace Technology Co., Ltd.), and 1000 g copper-lead-zinc mixed concentrate was taken to corundum dry pot and put into the resistance furnace. The temperature of the resistance furnace was set to 1200 °C in oxygen atmosphere. When the temperature of the resistance furnace rose to the set temperature, timing started. After roasting for 3 hours, the oxygen supply was stopped, and the power supply of the resistance furnace was turned off. Meanwhile, the oxygen-passing tube was pulled out, and the calcined product was taken out for cooling, manually crushing and grinding to -74 um for leaching”;

The dissolution experiments were conducted in a laboratory apparatus (DF-II digital display collector type magnetic stirrer, Jintan Shenglan Instrument Manufacturing Co., Ltd.)”;

“Zinc was tested by atomic absorption spectroscopy, and copper and lead were determined by volumetric method”.

Besides, the description of oxidizing roasting experiment has been moved to section 3 “Results and discussion”, and more detailed description of pyrometallurgical part has been provided in the revised manuscript according to the Reviewer’s suggestion.

2) What was the composition of raw ore?

Response to 2): Thank you for your careful review. The composition of raw ore contains 0.53% Cu, 1.29% Pb and 0.54% Zn, and the Au and Ag contents are 0.28 g/t and 23.6 g/t, respectively.

3) What was the mass balance of the oxidizing roasting process? Was there any volatile content which was lost in the form of gas/dust.

Response to 3): Thank you for your careful review. In the oxidizing roasting process, oxygen was added, and S in the ore samples was lost in the form of gas.

2. Results and Discussion

1) Temperature effect - Could you explain why zinc leaching extend/yield at 65℃ was significantly lower than at other temperatures? Could you provide the value of the measurement uncertainty/error, i.e. in the form of error bars?

Response to 1): Thank you for your careful review and valuable questions. Zinc leaching extend at 65℃ was significantly lower than at other temperatures, and this phenomenon was also reported by Kim et al. [10], who investigated the effect of temperature on the selective leaching Pb and Cu from secondary lead smelting residues. They showed that the leaching extent of Pb and Cu dramatically decreased with temperature due to PbSO4 formation at higher temperature.

[10] Kim E., Horckmans L., Spooren J., Vrancken K.C., Quaghebeur M., Broos K.; Selective leaching of Pb, Cu, Ni and Zn from secondary lead smelting residues[J]. Hydrometallurgy, 2017, 169:372-381.

2) Different resolutions of XRD patterns in Figure 1 and 3 make comparison difficult. 

Response to 2): We are very sorry for making you misunderstood. In this paper, Figure 1 is the XRD pattern of the material before leaching, and Figure 3 is the XRD pattern of the leaching residues at different leaching temperatures. The comparison of Figure 1 and Figure 3 shows that the composition of the material in the figure is different.

3) H2SO4 dosage effect - Could you explain why leaching extend of Cu and Zn significantly decrease when H2SO4 dosage is increased from 828 to 920 kg/t. Could you provide the concentration of Zn and Cu in the solution after leaching? Why you provide the concentration of H2SO4 in kg/t. It is usually presented in wt.% or M (molar) in the scientific publications.

Response to 3): Thank you for your careful review and valuable suggestions. The leaching extent of Cu and Zn significantly decrease when H2SO4 dosage is increased from 828 to 920 kg/t, combining with the XRD pattern analysis, which may be due to that a large amount of PbSO4 covers the surface of copper and zinc, hindering the contact between leaching agent and target elements, and reduceing the leaching rate of copper and zinc.

In the actual ore leaching test, the unit of sulfuric acid is generally expressed in kg/t, however, the unit of sulfuric acid is usually presented in wt.% or M (molar) in the pure mineral test. For example, Bai et al [18] also provided the concentration of H2SO4 in kg/t in the leaching of copper from refractory copper oxide ores.

[18] Bai X., Wen S. M., Feng Q. C., Liu J., Lin Y. L.; Utilization of high-gradient magnetic separation–secondary grinding–leaching to improve the copper recovery from refractory copper oxide ores[J]. Minerals Engineering, 2019, 136:77-80.

3. Others: English need to be improved within the text. There are some mistypes, e.g. in the all Figures Znic should be replaced by Zinc, (line 105) Twenty grams of calcine were placed, etc.

Response to 3: We are very sorry for our negligence of writing. We have made correction in the revised manuscript according to the Reviewer’s comments, and we have done our best to improve the English expression of the whole manuscript to make the paper more readable.

Once again, thanks very much for your comments and suggestions. We appreciate for your warm work earnestly, and hope that the correction will meet with approval.

Round  2

Reviewer 1 Report

The authors have adequately revised the manuscript to address my comments.

Reviewer 2 Report

The revised manuscript is acceptable for publication.

Reviewer 3 Report

Dear Authors,

I received answerws to my remarks and they were also included in the revised version of the manuscript. However, the information about analytical equipment is still missing. Information about equipment used for XRD, AAS, etc. analysis should be provided in Materials and Methods section.

Therefore, I recommed to accept the manuscript if the missing information will be provided.